

# Cost-effort analysis of Baited Remote Underwater Video (BRUV) and environmental DNA (eDNA) in monitoring marine ecological communities

Alice J. Clark[1], Sophie R. Atkinson[1], Valentina Scarponi[1], Tim Cane[2], Nathan R. Geraldi[3], Ian W. Hendy[4], J. Reuben Shipway[5] and Mika Peck[1]

[1] Department of Ecology & Evolution, School of Life Sciences, University of Sussex, Brighton, United Kingdom
[2] Department of Geography, University of Sussex, Brighton, United Kingdom
[3] NatureMetrics, Guildford, United Kingdom
[4] School of Biological Science, University of Portsmouth, Portsmouth, United Kingdom
[5] School of Biological and Marine Sciences, University of Plymouth, Plymouth, United Kingdom

Corresponding author
Alice J. Clark, ac831@sussex.ac.uk

## ABSTRACT

Monitoring the diversity and distribution of species in an ecosystem is essential to assess the success of restoration strategies. Implementing biomonitoring methods, which provide a comprehensive assessment of species diversity and mitigate biases in data collection, holds significant importance in biodiversity research. Additionally, ensuring that these methods are cost-efficient and require minimal effort is crucial for effective environmental monitoring. In this study we compare the efficiency of species detection, the cost and the effort of two non-destructive sampling techniques: Baited Remote Underwater Video (BRUV) and environmental DNA (eDNA) metabarcoding to survey marine vertebrate species. Comparisons were conducted along the Sussex coast upon the introduction of the Nearshore Trawling Byelaw. This Byelaw aims to boost the recovery of the dense kelp beds and the associated biodiversity that existed in the 1980s. We show that overall BRUV surveys are more affordable than eDNA, however, eDNA detects almost three times as many species as BRUV. eDNA and BRUV surveys are comparable in terms of effort required for each method, unless eDNA analysis is carried out externally, in which case eDNA requires less effort for the lead researchers. Furthermore, we show that increased eDNA replication yields more informative results on community structure. We found that using both methods in conjunction provides a more complete view of biodiversity, with BRUV data supplementing eDNA monitoring by recording species missed by eDNA and by providing additional environmental and life history metrics. The results from this study will serve as a baseline of the marine vertebrate community in Sussex Bay allowing future biodiversity monitoring research projects to understand community structure as the ecosystem recovers following the removal of trawling fishing pressure. Although this study was regional, the findings presented herein have relevance to marine biodiversity and conservation monitoring programs around the globe.

## INTRODUCTION

Ecosystems worldwide are increasingly threatened by anthropogenic activities, and marine ecosystems are no exception (*Luypaert et al., 2020*; *Bugnot et al., 2020*). Coastal ecosystems, in particular, suffer from a range of direct and indirect human stressors (*Lotze et al., 2006*; *Beck & Airoldi, 2007*; *Wernberg et al., 2011*; *Andersen et al., 2020*) including overfishing (*Jackson et al., 2001*; *Coll et al., 2008*; *Link & Watson, 2019*), habitat degradation (*Layton et al., 2020*), pollution (*Steneck et al., 2002*; *Lu et al., 2022*) and climate change (*Steneck et al., 2002*; *Smale et al., 2013*; *Brodie et al., 2014*; *He & Silliman, 2019*). These stressors are detrimental to biodiversity, leading to population declines, loss of genetic diversity and potentially result in species extinctions (*Butchart et al., 2010*; *Díaz et al., 2019*; *Yan et al., 2021*; *O'Hara, Frazier & Halpern, 2021*). Furthermore, the loss of biodiversity will likely affect the ocean's ability to provide food, protect livelihoods and recover from environmental stressors such as storms (*Cardinale et al., 2012*; *Crowe & Rotherham, 2019*; *Funge-Smith & Bennett, 2019*; *Talukder et al., 2022*). Fortunately, policy strategies to protect and restore marine biodiversity are emerging (*Duarte et al., 2020*). Examples of these strategies include Sustainable Development Goal 14 (SDG14) of the United Nations ("conserve and sustainably use the oceans, sea and marine resources for sustainable development"), the EU Biodiversity Strategy, and the Convention on Biological Diversity's proposal for 30% of the global ocean to become marine protected areas (MPAs) by 2030. Monitoring the diversity and distribution of species in an ecosystem is essential to assess the success of conservation and restoration strategies such as these mentioned above. However, the methods for monitoring marine and aquatic ecosystems can be inefficient and biased (*e.g.*, diver operated video; *Lindfield et al., 2014*), harmful to the environment and biodiversity (*e.g.*, bottom trawls and seine nets; *Kelly et al., 2014*), and time-consuming and expensive (*Sassoubre et al., 2016*). Therefore, choosing an efficient and cost-effective sampling method to establish baselines and monitor biodiversity is an important consideration in an ecological study to provide guidance for management decisions (*Thomas, 1996*; *Rotherham et al., 2007*; *Stat et al., 2019*).

Baited Remote Underwater Video (BRUV; *Harvey & Shortis, 1995*) is an increasingly common, effective, non-invasive and non-destructive method for sampling marine biodiversity (*Langlois et al., 2020*). The use of BRUV systems is particularly favoured over extractive sampling methods, such as trawling, in protected areas or areas where the goal is to enhance and protect biodiversity (*Langlois et al., 2020*). They provide a 40% more efficient approach to recording species counts than diver video transects (*Langlois et al., 2010*; *Watson et al., 2010*). Additionally, BRUV offers a permanent sampling record that can be reviewed to reduce interobserver variability (*Cappo et al., 2009*), provides data on habitat types (*Bennett et al., 2016*) and can be deployed in deep or heavily structured ecosystems that are hard to sample using other methods such as seines or trawls (*Esmaeili et al., 2021*). As multiple BRUV systems can be deployed at one time, it is also a time-efficient survey method (*Langlois, Harvey & Meeuwig, 2012*). BRUV can provide relative measures of species richness and abundance for a range of species and in a diverse range of conditions and habitats (*Cappo, Harvey & Shortis, 2006*). Stereo-BRUV systems also have

the advantage of being used to measure the body size of fish (*Harvey, Fletcher & Shortis, 2001*). Fish size can be used as a proxy for biomass as an essential metric for biodiversity and conservation as well as for fisheries management reporting (*Langlois, Harvey & Meeuwig, 2012*).

However, some studies have highlighted the limitations of BRUV. Species ID can be challenging in turbid water environments due to low visibility (*Harvey, Fletcher & Shortis, 2002*; *Cappo, Speare & De'ath, 2004a*; *Cappo, Speare & De'Ath, 2004b*; *Unsworth et al., 2014*) and there can be overrepresentation of apex predators due to biases associated with bait choice (*Harvey et al., 2007*; *Coghlan et al., 2017*; *Jones et al., 2020*; *Shah Esmaeili et al., 2022*). In addition, the scent of the bait will draw species from other areas that may not necessarily be local to the sites sampled, thus the sampling area is largely unknown (*Schramm et al., 2020*). Cryptic and sedentary species are also likely to be under-represented when using BRUV (*Watson et al., 2005*; *Stobart et al., 2007*; *Harvey et al., 2007*; *Stat et al., 2019*; *Shah Esmaeili et al., 2022*). Finally, footage analysis can be labour-intensive, time-consuming and costly (*Ditria et al., 2021*).

Metabarcoding of environmental DNA (eDNA) for biomonitoring has become increasingly popular in marine ecosystems. It is a non-invasive technique that removes the need for extensive taxonomic expertise to identify species typically required by traditional methods (*Rees et al., 2014*; *Smart et al., 2015*; *Evans et al., 2017*). eDNA is genetic material shed by organisms in the environment as skin cells, scales, faeces and other excrements (*Creer et al., 2016*). DNA will degrade over time but remains in the environment long enough for the presence of the organisms to be detected without being directly observed or caught (*Collins et al., 2018*). One of the main advantages of using eDNA-based assessments for biomonitoring is the ability to assess whole communities at once and detect rare and cryptic species, as well as undetected invasives (*Bohmann et al., 2014*; *Deiner et al., 2017*). eDNA-based surveys can be more sensitive and therefore less likely to underestimate species richness in the same way that most traditional fishery assessments do and have the potential to be cheaper than traditional techniques (*Willis, 2001*; *Port et al., 2016*). Another key advantage of using eDNA is the simplicity of sample collection: only relatively small volumes of water are needed and the filtering process is simple, requiring very little training, expertise and time in the field (*Miya et al., 2016*).

Despite the many advantages of eDNA-based monitoring, it also has limitations. Several factors can influence the detectability of eDNA in the environment, leading to either false negatives (failed detection of species present in the area) or false positives (detection of species not present in sampled area). Detection probability can be influenced by biotic and abiotic factors, including species-specific eDNA generation and degradation (*Deiner et al., 2017*), linked to body size, life history stage, diet and migration (*Sassoubre et al., 2016*; *Stewart, 2019*; *Rourke et al., 2022*). Additionally, contamination of the sample is possible from sample collection through to processing, affecting the accuracy of findings (*Goldberg et al., 2016*). Another limitation of eDNA is that reference databases used to translate the operational taxonomic units (OTUs) obtained from the DNA processing to taxa, are still incomplete, especially for species found in parts of the world where less research has been carried out (*Stoeckle, Mishu & Charlop-Powers, 2020*; *Schenekar et al., 2020*). The transport

of eDNA by ocean currents (*Goldberg et al., 2016*; *Andruszkiewicz Allan et al., 2021*) and its rate of decay due to UV strength, pH and water temperature can also affect detection probability (*Sassoubre et al., 2016*). Nonetheless, eDNA signals remain relatively accurate to the sample site (*Kelly, Gallego & Jacobs-Palmer, 2018*) because eDNA degrades relatively rapidly in marine environments (from a few hours to a couple of days; (*Collins et al., 2018*; *Murakami et al., 2019*). Thus, the use of eDNA for biomonitoring marine environments promises to be powerful. However, eDNA is not yet able to provide information about size, life stages, or sex ratios (*Mynott, 2020*), thus it is still essential to use it in combination with another sampling method, such as BRUV, to collect this type of data.

eDNA metabarcoding and BRUV surveys offer strong potential to monitor inshore coastal habitats. Assessing how these biomonitoring methods perform compared to one another in terms of species detection sensitivity and relative to cost and effort is important, as it may influence future monitoring studies. Recent studies comparing eDNA and underwater videos for biomonitoring have found that eDNA detected a greater percentage of total genera alone than underwater videos (*Stat et al., 2019*; *Mirimin et al., 2021*). However, these comparative studies, along with others, have concluded that using both methods in conjunction improved the overall species richness detected (*Stat et al., 2017*; *West et al., 2020*; *Cole et al., 2022*; *Mirimin et al., 2021*; *Gold et al., 2023*). While many studies have used both BRUV and eDNA, few studies have explored the cost-effectiveness of each method. This is an important gap which needs to be addressed as it may aid recovery projects to focus on a particular method based on the cost or effort it requires.

The objectives of this study were to: (1) compare species assemblage metrics obtained using BRUV and eDNA; (2) compare the sensitivity of two eDNA metabarcoding primers (MiFish 12S and Valsecchi 16S); (3) investigate the importance of eDNA replication and (4) compare the cost and effort required of both survey techniques for detecting the presence of marine vertebrate species.

## MATERIALS & METHODS

### Study area

Sussex Bay, on the south coast of the UK, is an ideal site to test these biomonitoring methods. Historically, the coast of Sussex harboured dense kelp beds, which provided shelter and nursery grounds for many species while also protecting the coastlines from storm damage and sequestering $CO_2$ from the atmosphere. However, the "Great Storm" of 1987 damaged these kelp ecosystems. Trawling, and subsequent storms, along with increased sedimentation and rising water temperatures, attributed to the loss of 96% of the Sussex kelp beds that were present in the 1980s (*Williams & Davies, 2019*). In 2021, the Sussex Inshore Fisheries and Conservation Authority (SxIFCA) introduced the Sussex Nearshore Trawling Byelaw, preventing trawling activity along 304 km$^2$ of the Sussex Coast. This new Byelaw aims to provide an opportunity for kelp and native fish to recover leading to an overall healthier coastal ecosystem. The trawling ban, and the expected ecosystem recovery, also offer an opportunity for the exploration of new techniques in biomonitoring.

We examined the presence of marine vertebrate biodiversity along the Sussex coast at 28 different sites between Selsey Bill (50°43.325′N, 0°46.040′W) and Shoreham-by-Sea

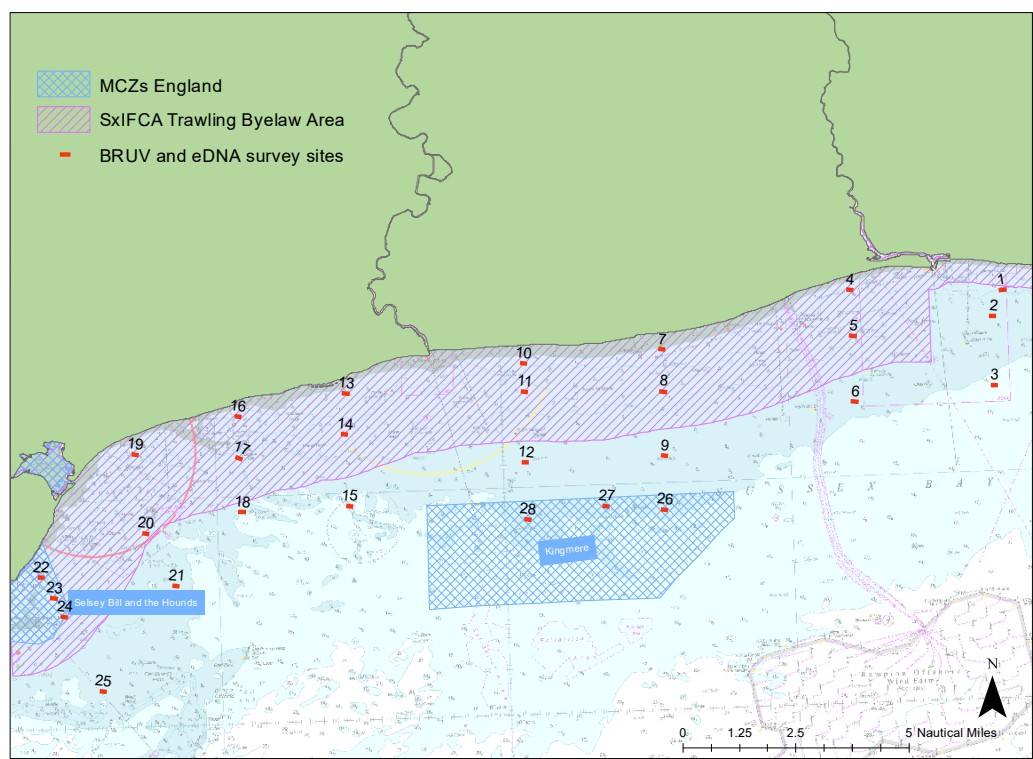

**Figure 1  Site map.** Map of study region showing the 28 sampled sites within Sussex Bay. A separate deployment of 3 BRUV systems was undertaken in Swanage in a kelp-dominated ecosystem for reference. The red dots and black numbers correspond to the sites sampled. Sites within the pink hashed area are within the Trawling Byelaw Area, sites within blue hashed areas are in Marine Conservation Zones (MCZs) and sites outside hashed areas are the sites in areas which have no protection.

(50°48.981′N, 0°12.265′W) (Fig. 1). The sites were chosen to match the towed video transects deployed by SxIFCA, to complement their habitat data (*Mallinson & Yesson, 2020*). Of these sites, 10 were outside the trawling exclusion zone, 12 were within the trawling exclusion zone and six of them were within Marine Conservation Zones (Kingmere and Selsey Bill and the Hounds). An additional site in Swanage (50° 44.272′N, 0°29.077′W) was also sampled to provide comparative information on a kelp-dominated ecosystem. The samples were collected between the 5th and 21st of July 2021 during daylight hours, between 8 am and 5 pm.

## Baited remote underwater video (BRUV)

The stereo-BRUV systems used in this study were based on those used by *Harvey, Fletcher & Shortis (2002)* and *Watson et al. (2005)*. Each BRUV system was equipped with two GoPro HERO 8 cameras facing the bait canister, for which the settings were standardised to: 1080p, linear and 30 FPS. A third camera was placed to face the back of the rig and set to record a time-lapse of the surrounding habitat, taking a picture every minute. The bait canister on each of the BRUV systems was filled with one semi-thawed and one frozen Atlantic horse mackerel (*Trachurus trachurus*), which had slits cut through them to

maximise the dispersal of their scent plume. A mixed leaded and unleaded (20 m long, 10 mm in diameter) line was attached *via* a floating bridal to each BRUV system to prevent the rope from obstructing the cameras' field of view. Two A0 buoys and one A2 buoy were attached at the end of each rope. Three BRUV systems were deployed in succession by boat at each of the 29 sites, 150 m apart from one another and left to film the sea floor for up to 75 min. The date and the time of deployment were recorded at each site, as well as GPS coordinates (latitude and longitude).

The video images were analysed using footage from the right-hand camera, only using the left-hand camera in case the right camera failed or was obstructed by seaweed. Video analysis began 5 min after the BRUV hit the seafloor to allow the substrate to settle. To standardise the length of footage analysed we removed the species identified from the first and last GoPro clips, resulting in a standardised survey interval of 47 min and 8 s of footage. This was considered appropriate as the majority of species usually appear within the first 40–60 min of deployment (*Unsworth et al., 2014*). Fish and marine vertebrates were observed and identified to the lowest taxonomic level possible. The conservative measure of maximum number observed at a point in time (MaxN) was recorded for each species, to avoid double counting individuals (*Priede et al., 1994*; *Cappo, Speare & De'ath, 2004a*; *Cappo, Speare & De'Ath, 2004b*; *Langlois et al., 2010*).

## eDNA sampling

Three eDNA samples were collected at each of the 29 sites while the BRUV rigs were deployed. A 2L Kemmerer sampler was used to collect water samples. The water sampler was sanitised and rinsed in the water body between samples to avoid contamination from previous sites. The sampler was opened out of the water and then lowered into the water column from the boat. Samples were taken approximately one metre above the bottom of the seafloor to target the demersal layer, where eDNA is thought to be most concentrated (*Mynott, 2020*). When the sampler reached the desired depth, a messenger weight was released to operate the closing mechanism and force the sampler to shut with seawater trapped inside. To minimise eDNA contamination and degradation, each sample was filtered immediately on the boat using a peristaltic pump attachment on a Bosch drill (Vampire Sampler) and a Marpene Suction Hose. Two litres of water were passed through an encapsulated PES disk filter (0.8 µm mesh, five cm diameter filter) with a 5 µm glass fibre prefilter (NatureMetrics, UK), a salt and detergent-based lysis solution (Longmire's buffer) was added to the filter to preserve the eDNA. The filters were sent to NatureMetrics for further processing and analysis.

## eDNA analysis

The eDNA extraction and analysis followed the same methods as those described in *Alfaro-Cordova et al. (2022)* with some slight modifications described below:

A DNeasy Blood and Tissue Kit (Qiagen, Hilden, Germany) was used to extract DNA from the filters. Each batch of extractions included an extraction blank. A DNeasy PowerClean Pro Cleanup Kit (Qiagen) was used to remove PCR inhibitors. DNA within samples was measured with a Qubit 3.0 fluorometer (Thermo Fisher Scientific, Waltham, MA, USA).

Amplification and indexing included a two-step PCR process with primer pairs amplifying a section of the 12S ribosomal RNA (teleost fish; *Miya et al., 2015*) and 16S ribosomal RNA (vertebrates; *Valsecchi et al., 2020*). Primer pairs were the MiFish 12S loci (Forward-GYYGGTAAAMYTCGTGCCAGC and Reverse-CATAGYGGGGTATCTAATCCCRGTTTG) and the Valsecchi 16S loci (Forward-AGACGAGAAGACCCTRTG, Reverse-GGATTGCGCTGTTATCCC). The initial PCR included multiple PCR replicates for each sample. The PCR step for the MiFish assay consisted of 1X Phusion Green PCR Master Mix (Thermo Fisher Scientific), 0.4 µM of the forward and reverse primer, 3% DMSO, 0.9 µl of sample DNA, 1.5 mM of $MgCl_2$ (Thermo Fisher Scientific), 0.6 mg/ml of BSA (Thermo Fisher Scientific), and molecular grade $H_2O$ (Thermo Fisher Scientific). The PCR mixture had a final volume of 8 µl. The PCR conditions in *Miya et al. (2015)* were used for the MiFish assay with the exception of a touchdown annealing step of 10 cycles PCR ($-0.5$ °C per cycle) starting at 69 °C, then 25 cycles of 72 °C for 15 s.

The Valsecchi 16S amplification mixture had a volume of 8 µl and contained 1X DreamTaq Green PCR Master Mix (Thermo Fisher Scientific), 0.54 µM of the forward and reverse primer, 0.9 µl of sample DNA, and molecular grade $H_2O$ (Thermo Fisher Scientific). The Vasecchi 16S assay PCR conditions had an initial denaturation at 94 °C for 4 min followed by 38 cycles. Each cycle involved denaturation at 95 °C, for 30 s, followed by four 30 s touchdown annealing steps ($\pm$ 1 °C, per cycle) starting at 60 °C, then 72 °C, for 40 s; and a final elongation step at 72 °C, for 5 min. Amplicons generated using these primer sets are hereafter referred to as MiFish 12S and Valsecchi 16S.

The libraries for both assays were processed the same way. Sequences were demultiplexed with bcl2fastq and then two different pathways were followed to calculate ZOTUs (zero-radius OTUs). Reads were binned using USEARCH with the paired-end FASTQ reads for every sample (*Edgar, 2010*). The forward and reverse primers were removed using cutadapt (*Martin, 2011*). Sequences were filtered based on length for the assay. These sequences were then quality filtered and dereplicated using USEARCH. Unique sequences from all samples were denoised using the UNOISE program (*Edgar, 2016*).

Taxonomy was assigned using BLASTN (*Altschul et al., 1990*; *Camacho et al., 2009*) with the NCBI nucleotide database (NCBI nt; downloaded on 28-09-2021). Minimum similarity thresholds of 99%, 97% and 95% for species, genus, and higher-level assignments were used respectively. Final taxonomy was from GBIF and we removed sequences that were not assigned a kingdom. OTUs with low abundance were also removed. This included OTUs with less than 0.02 and 0.025% of all sequences for MiFish 12S and Valsecchi 16S assays, respectively. Identified species were cross-referenced with fish distribution maps from FishBase (*Froese & Pauly, 2024*), two taxa detected had no occurrence in the UK and were removed from the analysis.

## Environmental and spatial descriptors

Experimental design consisted of six factors: Treatment (five levels: inside trawling exclusion zone–12 sites, outside trawling exclusion zone–10 sites, Marine Conservation Zone Kingmere–three sites, Marine Conservation Zone Selsey Bill and the Hounds–three

sites and a control site with healthy kelp in Swanage–one site), Site (28 sites following SxIFCA towed transects and the Swanage site), Macroalgae percentage cover (estimated from habitat camera on the BRUV), Biotype (6 levels: Gravel-cobbles, Sand, Mixed sediment, Rocky reef, Cobbles and pebbles, kelp rock), Tidal Stream (calculated using the Admiralty Chart and Reeds Nautical Almanac 2021; *Towler & Fishwick, 2020*) and Depth (in metres, based on the boat's sonar system). The coordinates were converted to Easting and Northing (APPENDIX: Table 1).

## Data pre-processing

In total, 87 BRUV deployments were carried out and 87 eDNA samples were collected in this study. Data from the three BRUV systems at each site were pooled to avoid pseudo-replication, resulting in 29 samples. The MiFish assay was only used for the first 29 replicates, due to funding limitations. Thus, for the statistical analysis to compare sensitivity of BRUV and eDNA surveys, only the first replicates from each site were statistically analysed using the results from both the MiFish 12S and the Valsecchi 16S assays. Site 9 failed to amplify, likely due to low concentrations of eDNA in the sample, perhaps due to errors made during the sampling process or during extraction. Therefore, Site 9 was removed from both the BRUV and eDNA datasets, resulting in a total of 28 sites analysed. Sites 4, 5, 9, 10, 14, 16, 20 and 28 had one or two samples out of the three replicates from each site that amplified. The lack of amplification was likely caused by low concentrations of DNA, although we cannot rule out PCR inhibition which is less likely given that some replicates amplified. For these reasons, the statistical analyses investigating the benefit of replication, using the Valsecchi 16S assay results, included 21 of the Sussex sites (for which we had 3 successful replicates at each site) and compared the benefit of taking 63 eDNA replicates, 42 replicates and 21 replicates. The eDNA OTU read counts were converted to presence-absence data per sample. Although recent work has demonstrated that eDNA read counts can be used as a proxy for abundance, a stronger correlation has been found in controlled experiments than in the field (*Yates, Fraser & Derry, 2019*), likely due to the uncertainties linked to the field estimation of organism abundance by the conventional sampling method (*Di Muri et al., 2020*). Thus, as neither of the biomonitoring methods used give accurate abundance estimates, we focussed on presence-absence data only and transformed BRUV data from MaxN to presence-absence data.

Organisms that were not identified to genus level were removed from both eDNA and BRUV datasets. Only five species from the eDNA dataset were identified to genus level and no further. All taxa detected on the BRUV footage were identified to species level. Freshwater species and other non-target species detected (*e.g.*, mallard, *Anas platyrhynchos*) were also removed from the eDNA dataset. Atlantic horse mackerel (*T. trachurus*) was used as bait for the BRUV and therefore was also removed from the eDNA dataset as there was no way of determining whether the DNA from this species was from naturally occurring individuals or from the bait.

## Data analysis

We analysed the data in the programming language R (v. 4.1.2) in RStudio (*R Core Team, 2023*). To identify any underlying spatial autocorrelation structure in our ecological data

we used distance-based Moran's eigenvector mapping (dbMEM; *Borcard & Legendre, 2002*; *Borcard et al., 2004*; *Peres-Neto et al., 2006*; *Legendre, Fortin & Borcard, 2015*) with the *adespatial* package in R (*Dray et al., 2023*).

For each method, sample coverage and the fraction of the total incidence probabilities of the detected species for a set of sampling units were estimated from rarefaction and extrapolated models for species richness (Hill number $q = 0$) using the *iNEXT* package (*Hsieh, Ma & Chao, 2016*). This was also used to assess the differences between the number of replicates taken at each site.

The two primers were compared using multivariate analysis and the difference in species assemblage detected by each primer was assessed using the *manyglm* function from the *mvabund* package (*Wang et al., 2022*). The homogeneity of group dispersions on Jaccard dissimilarity indices between the primers was analysed using the *betadisper* function from *vegan* (*Oksanen et al., 2022*).

To assess how species composition varied along the environmental gradient for both survey techniques a Canonical Correspondence Analysis (CCA) was performed with environmental variables (tidal stream, depth, macroalgal cover, biotype) and community composition. Log transformation was applied to the numerical environmental variables included in the analysis to normalise the data in order to improve estimation efficiency and interpretation. The first two axes of the CCA were plotted together with environmental variables (*Oksanen et al., 2022*). The most parsimonious model of the CCA was found using forward and backward selection (*ordistep* function). The *mvabund* package was used to determine which treatment types were significantly different to one another in terms of species composition (*Wang et al., 2022*). Homogeneity of dispersions on Jaccard dissimilarity indices between the two sampling methods was tested using the *betadisper* function (*Oksanen et al., 2022*). The same analyses were conducted on the eDNA sample replication analysis.

## Cost and effort comparison

The average yearly cost of sampling the 28 Sussex sites over 5 years was calculated to compare cost-effectiveness of each biomonitoring method (APPENDIX: Table 2). The cost was calculated using prices from 2021 and assuming only one eDNA sample was taken per site and three BRUV rigs deployed per site. As eDNA can be widely dispersed by tidal currents, we expect the samples to reflect the species from a wider geographical area than just the location from which the sample was taken. It is for this reason that we chose to compare 1 eDNA sample to the three BRUV deployments spread over 300 m at the same site. The BRUV costs included: building the BRUV rigs, the GoPro Hero8 cameras, the bait, the boat hire and the labour cost for video analysis. As the BRUV rigs were built the first year and reused for the following 4 years, the first year cost more than subsequent years. The eDNA costs covered: the eDNA kits from NatureMetrics, the Kemmerer sampler, Vampire suction drill hire, the boat hire and eDNA extraction and analysis by NatureMetrics. Again, the first year was more expensive than subsequent years as we assume purchased equipment will be reused. We also calculated the cost of carrying out the analysis in-house as opposed to outsourcing the analysis. Here we included the cost
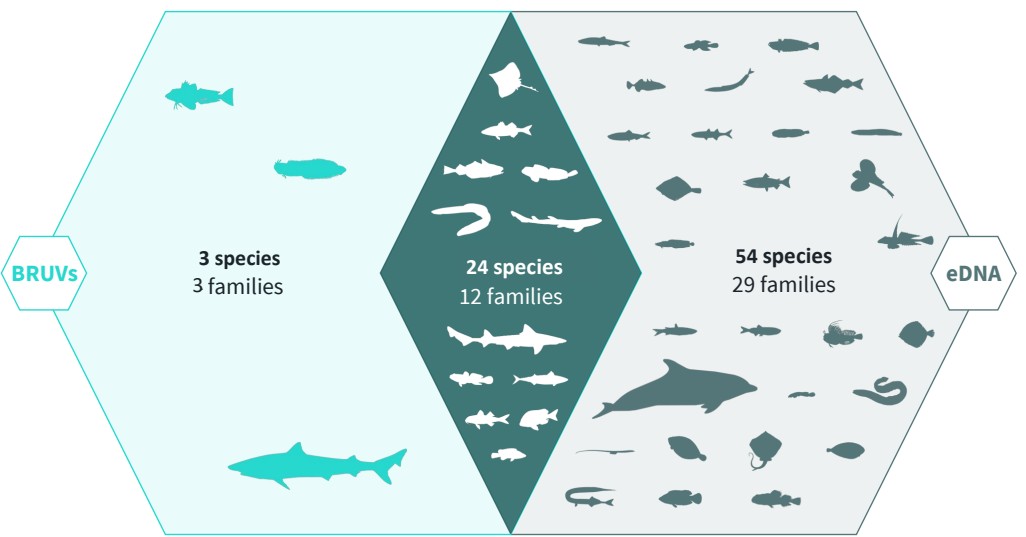

**Figure 2  Venn diagram of eDNA and BRUV species detections.** Environmental DNA surveys captured the majority (78/81) of the species detected by both surveys, failing to identify only three species detected by the BRUV. eDNA identified 54 species that were missed by the BRUV surveys. In contrast, BRUV surveys identified 27/81 species. Both methods identified the same 24 species belonging to 12 families. Infographic was designed by Alice Clark in collaboration with NatureMetrics.

of laboratory consumables (*e.g.*, reagents, DNA extraction kits, tubes) in our estimates, but excluded general purpose laboratory equipment (*e.g.*, pipettes, centrifuges, thermal cycler). Labour costs for the analysis were also included in the calculation. We recognise that a different level of expertise is required for BRUV video analysis and eDNA in-house analysis, therefore this was accounted for in the labour costs of each method. The time taken to carry out the fieldwork and the analysis for the 28 samples was calculated to estimate the effort required by each method.

# RESULTS

## Species diversity

Across 28 survey sites we recorded a total of 81 taxa. Using the BRUV imagery 27 species belonging to 15 families were identified, whereas eDNA analysis detected a total of 78 species belonging to 41 families (Fig. 2, Fig. 3). Both methods detected the same 24 species belonging to 12 families. BRUV identified 3 species that were not detected using eDNA: Yarrel's blenny (*Chirolophis ascanii*), tub gurnard (*Chelidonichthys lucerna*) and tope shark (*Galeorhinus galeus*). In contrast, 54 species were detected using eDNA alone (APPENDIX: Table 3), of note is the European eel (*Anguilla anguilla*) which is classed as Critically Endangered on the IUCN RedList (*IUCN RedList, 2022*).

eDNA had higher sample coverage estimates (96.3%) than BRUV (94%) surveys (APPENDIX: Fig. 1). The estimated species richness ($q = 0$) was 26.9 ± 7.9 SE for the BRUV surveys and 102.3 ± 20.2 SE for the eDNA surveys. The average observed species richness per site was six times higher for samples collected by eDNA than for BRUV samples

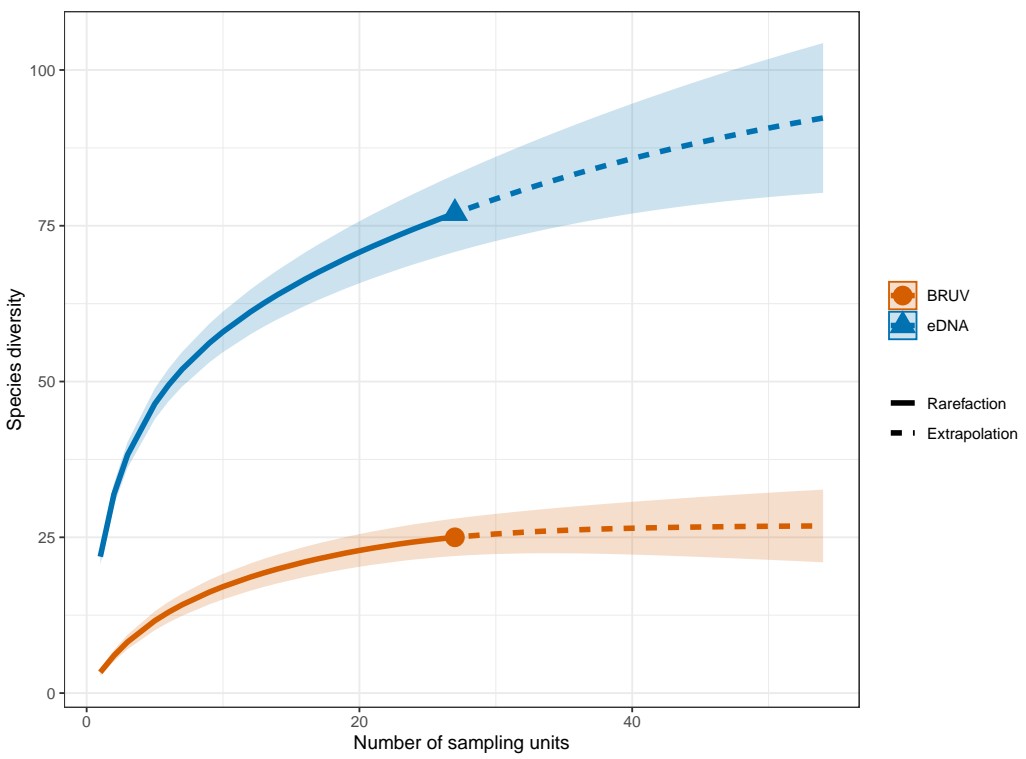

**Figure 3** **Sample-sized based rarefaction and extrapolation curves of fish and marine vertebrates detected using BRUV and eDNA surveys.** Overall, across the 27 sites in Sussex, the observed species richness for eDNA was 78, while BRUV detected 26 species. Shaded areas represent a 95% confidence interval.

(mean species richness per site: eDNA = 22 ±5.9 SD, BRUV = 3.3 ±2.1 SD). The first 3 most detected species by each method were different. For eDNA these were: black seabream (*Spondylisoma cantharus*, detected at 100% of sites), Atlantic mackerel (*Scomber scomber*, detected at 100% of sites) and European seabass (*Dicentrarchus labrax*, detected at 88.9% of sites), all of which were also detected by BRUV but fewer times. The three most detected species by BRUV surveys were: small spotted catshark (*Scyliorhinus canicula*, detected at 44.4% of sites), bib (*Trisopetrus luscus*, detected at 37% of sites) and conger eel (*Conger conger*, detected at 33.3% of sites), these were also detected by the eDNA surveys but less frequently.

### Primer comparison

Metabarcoding using the MiFish 12S primers detected a higher number of species ($N = 70$) than the Valsecchi 16S primers ($N = 58$). The MiFish 12S primers detected 35 unique species which were not picked up by Valsecchi 16S, and Valsecchi 16S detected 23 species missed by MiFish 12S. There was an overlap of 36.6% between the two primers. In terms of species assemblage, a significant difference was found between the two assays (*manyglm: df* = 1, dev = 375.7, wald value = 7.7, $p = 0.002$, APPENDIX: Fig. 2). The homogeneity of dispersions between the two primers was also significantly different (*betadisper: F* = 21.3, $p < 0.001$; APPENDIX: Fig. 3).

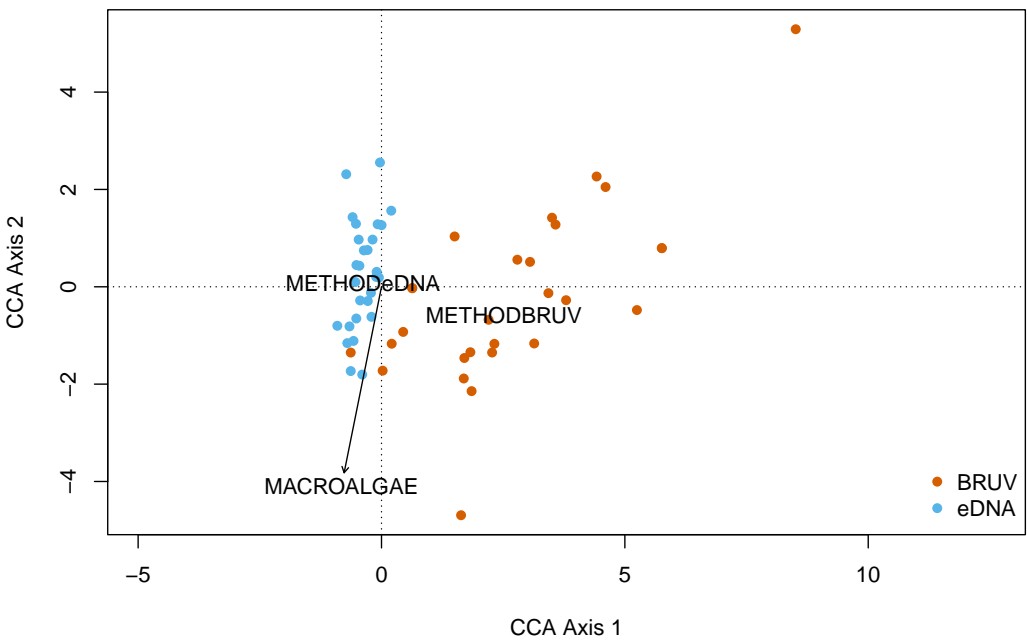

**Figure 4 Canonical correspondence analysis of eDNA *vs* BRUV.** Canonical correspondence analysis (CCA) showing sampled sites clustered by sampling method. eDNA sampled sites are in blue and BRUV sampled sites are in orange. The environmental variable macroalgal percentage cover is plotted as a black vector. The CCA revealed that species and sites and structured by macroalgal percentage cover ($\chi^2 = 0.25$, $F = 1.71$, $p = 0.008$) and sampling method ($\chi^2 = 0.34$, $F = 3.65$, $p = 0.001$).

## Community Structure

No significant spatial autocorrelation was detected by the global dbMEM analysis (truncation level $= 0.11$, $p = 1$). Analysing species composition at the 27 sites in Sussex and the control site in Swanage with CCA, yielded weak ties and hence should be considered with caution. The first two CCA axes were significant (CCA1: $\chi^2 = 0.33$, F(1, 53) $= 3.75$, $p = 0.001$; CCA2: $\chi^2 = 0.15$, F(1, 53) $= 1.63$, $p = 0.022$) and together explained 48% of the total variability. The CCA revealed that species and sites are structured by macroalgal percentage cover and sampling method (Fig. 4, CCA stepwise permutation selection, "macroalgae": $\chi^2 = 0.15$, F(1, 53) $= 1.71$, $p = 0.008$; "method": $\chi^2 = 0.34$, F(1, 53) $= 3.65$, $p = 0.001$, for the other variables: $p > 0.05$). The CCA demonstrated that BRUV and eDNA yielded different community structures (permutation test: F(1, 53) $= 3.6$, $p = 0.001$) and there was significant variation in the beta-dispersion between the two methods (*betadisper*: F(1, 54) $= 76.2$, $p = 6.65e-12$; APPENDIX: Fig. 4). A significant negative difference in multivariate dispersions suggests that eDNA samples have a lower multivariate dispersion compared the BRUV samples (Tukey test: diff: $-0.26$).

## eDNA sample replication

Differences in species diversity ($q = 0$) among replicate numbers (*i.e.*, one replicate per site–21 samples, two replicates per site–42 samples, three replicates per site–63 samples) at each site were observed (Fig. 5). In the one replicate per site assemblage (21 samples) the

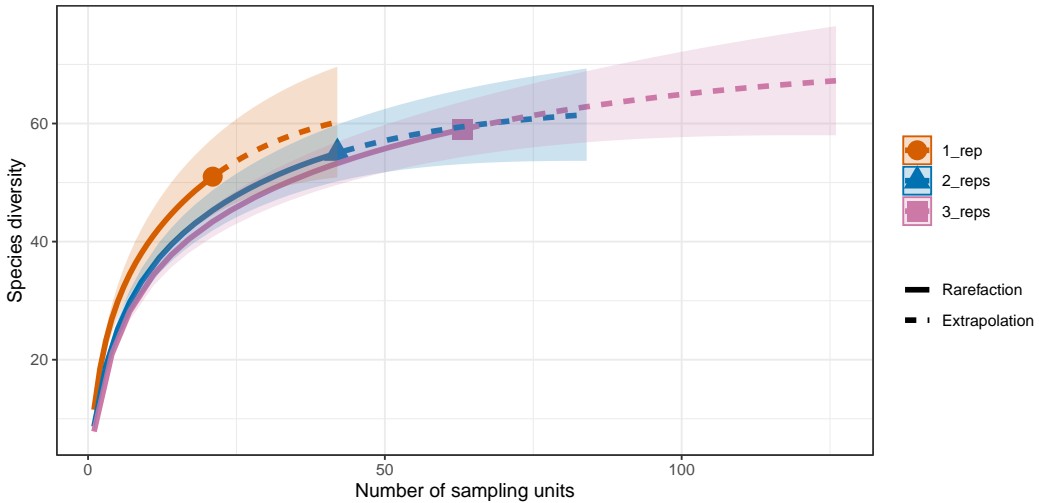

**Figure 5  Sample-sized based rarefaction and extrapolation curves of the Valsecchi 16S assay comparing the number of replicates taken at each Sussex site.** The observed species richness ($q = 0$) for 1 replicate per site (21 samples) was 51, for two replicates per site (42 samples) it was 55 and for 3 replicates per site (63 samples) it was 59. Shaded area represents the 95% confidence intervals.

observed species richness was 51 and the estimated species richness ($q = 0$) was 64.5 ±9.6 (SE). Extrapolation predicted a maximum richness of 60.3. The coverage-based estimate of diversity for 21 samples was 93.7% ($q = 0$). When two replicates were taken at each of the sites (42 samples) the observed species richness ($q = 0$) was 55 and the diversity estimate was 63.2 ±22.8 (SE). The extrapolation predicted a maximum richness of 59.3. For 42 samples, 96.5% of all the species jointly observed across all sampling strategies were detected. Finally, when three replicates were taken at each site the observed species richness was 59 while the diversity estimate was 71.1 ±22.8 (SE) ($q = 0$), with the extrapolation predicting a maximum richness of 64.1. With 63 sampling units, 98.4% of the species were detected.

The additional seven species detected in the three replicates per site compared to just one replicate per site were: tub gurnard (*C. lucerna*), Atlantic herring (*Clupea harengus*), tope shark (*G. galeus*), spotted ray (*Raja montagui*), common goby (*Pomatoschistus microps*), brill (*Scophthalmus rhombus*), poor cod (*Trisopterus minutus*) and topknot (*Zeugopterus punctatus*). Of note, two of these species, *G. galeus* and *C. lucerna*, are species that had been detected with the BRUV surveys and not with the first eDNA replicate taken at each site.

The global dbMEM analysis found no significant spatial structure within the dataset (truncation level = 0.11, $p = 0.136$). A CCA was conducted to explore the relationship between species composition and the environmental variables: depth, macroalgal cover and treatment. The total inertia in the dataset was 6.02. The first two axes collectively explain 14.33% of the total variation in the data. The CCA showed that the Sussex Bay fish and marine vertebrate community was structured by depth ($\chi^2 = 0.22$, F(1, 56) = 2.36, $p = 0.001$), macroalgae percentage cover ($\chi^2 = 0.15$, F(1, 56) = 1.66, $p = 0.016$) and treatment ($\chi^2 = 0.49$, F(4, 56) = 1.34, $p = 0.006$; Fig. 6). The MCZ Selsey Bill and the Hounds and

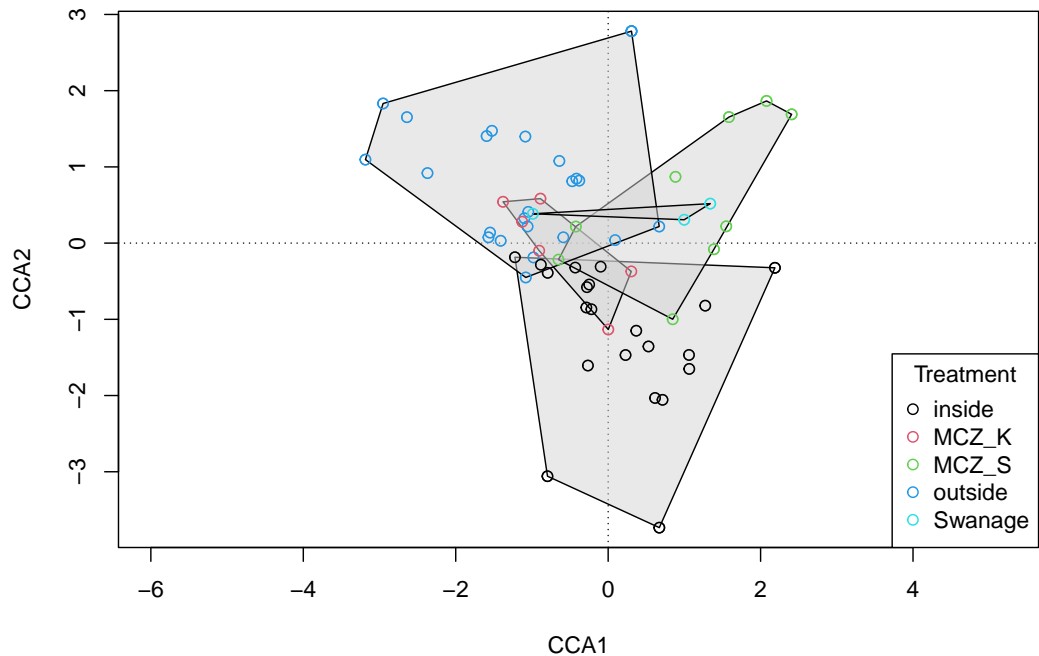

**Figure 6 Canonical correspondence analysis with additional replicates.** Canonical correspondence analysis (CCA) plot of the 21 Sussex sites and the Swanage control site with three eDNA replicates taken at each site. The sites are coloured according to treatment type (inside trawling exclusion zone, MCZ Kingmere, MCZ Selsey Bill and the Hounds, outside the trawling exclusion zone and the Swanage kelp control site). The CCA revealed that species and sites are structured by depth ($\chi^2 = 0.22$, $F = 2.36$, $p = 0.001$), macroalgal percentage cover ($\chi^2 = 0.15$, $F = 1.66$ $p = 0.006$) and treatment ($\chi^2 = 0.49$, $F = 1.34$, $p = 0.006$).

the sites outside the trawling exclusion zone were found to have a significantly different community composition to the other treatments (*manyglm*: Treatment MCZ Selsey Bill and the Hounds: *df* = 4, Dev = 312.9, wald value = 4.6, $p = 0.045$; Treatment outside: *df* = 4, Dev = 312.9, wald value = 5.3, $p = 0.007$; Treatment Swanage: *df* = 4, Dev = 312.9, wald value = 5.3, $p = 0.045$). The homogeneity of beta-distributions test revealed no significant difference between treatments (*betadisper*: $p > 0.05$).

## Cost and effort comparison

For the first five years of sampling, the average cost of making and deploying the BRUV systems at our 28 Sussex sites was £7,430 per year (Fig. 7). Assuming the cost of eDNA sequencing does not change within the next 5 years, the cost of taking one eDNA sample per site and getting them analysed externally would come to £14,072 on average per year for 5 years of sampling. The cost of eDNA sampling with the analysis conducted in-house would cost £12,250 per year. Cost per site for BRUV was £265, £502 for eDNA with external analysis and £440 for eDNA with in-house analysis. As BRUV surveys detected 27 species overall, the cost per species detected for BRUV was £275. eDNA detected 78 species, thus the external eDNA analysis was £180 per species detected, while eDNA with in-house analysis was £160 per species detected.
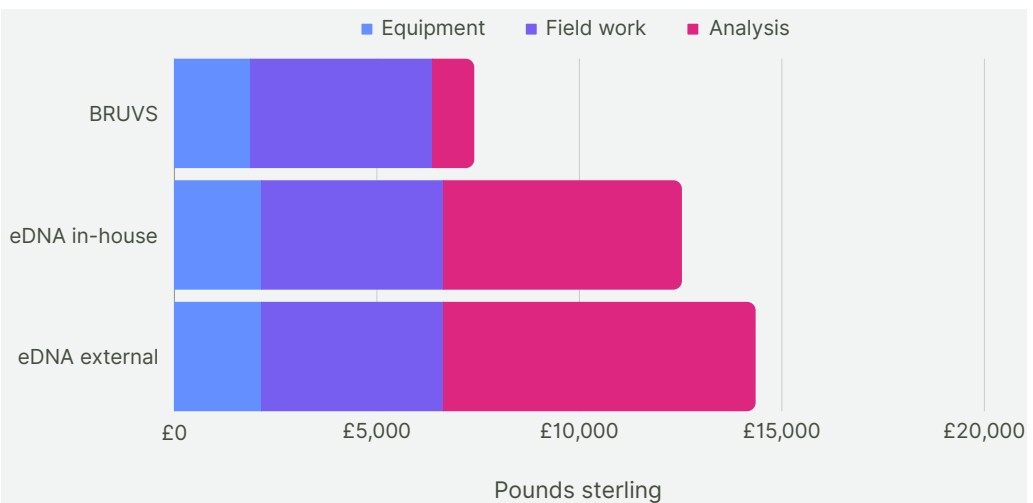

**Figure 7 Cost comparison.** Cost comparison of BRUV surveys, eDNA surveys with in-house sample analysis and eDNA surveys with external sample analysis. Costs were broken down into three categories: equipment, field work and analysis. Overall, the BRUV survey was the most affordable biomonitoring technique, and eDNA with external sample analysis was the most expensive.

Our fieldwork involved sampling three of our 28 sites a day over the span of 10 days. The time to take one eDNA sample was approximately 35 min. The process from sample collection to obtaining DNA sequence data for all sites extended to approximately 10 weeks, with 8 of these weeks spent waiting for the results to be returned. Had we opted for in-house eDNA analysis, the timeline would have been condensed to around 2 work weeks, resulting in 4-weeks from the initial data collection to a usable dataset. In contrast, we found that the total mean time to deploy, retrieve and analyse the footage from the BRUV systems for one site took 430 min. The analysis of the BRUV footage consumed around 95 h for all the sites, equivalent to two and half work weeks. Consequently, the duration from BRUV deployments to having a comprehensive dataset totalled 4 weeks. Thus, conducting BRUV sampling and eDNA sampling with in-house analysis takes roughly the same amount of time, whereas eDNA sampling with outsourced analysis takes longer.

## DISCUSSION

Non-destructive biomonitoring methods that accurately record biodiversity are vital for the future of marine recovery projects. Our findings align with those of previous studies and demonstrate the benefits of using two complementary non-destructive methods to assess biodiversity and community structure, while also shedding light on the biases inherent of both techniques and illustrating the cost and effort disparities between the two methods.

### Comparison between BRUV and eDNA surveys

The eDNA survey detected a higher species richness than the BRUV survey at the same sampling sites, demonstrating that eDNA was the more sensitive diversity monitoring technique. These results align with the findings of several previous studies (*Port et al., 2016*;

*Bakker et al., 2017*; *Valdivia-Carrillo et al., 2019*; *Stat et al., 2019*; *Jeunen et al., 2020*; *West et al., 2020*; *Afzali et al., 2021*; *Cole et al., 2022*; *Mirimin et al., 2021*; *Gold et al., 2023*). One species of note detected by eDNA but not by BRUV, was the European eel (*A. anguilla*), which is a Priority Species under the UK Post-2010 Biodiversity Framework and listed as Critically Endangered on the IUCN Red List of Threatened Species. BRUV surveys also detected species that the eDNA sampling missed, one of which was a tope shark (*G. galeus*), which is classed as Vulnerable by the IUCN. The detection of these Red List species by both methods underscores that the combined use of BRUV and eDNA surveys is crucial. Our results demonstrate that using only one method risks overlooking important protected species as well as key predators, which can be bioindicators of the health of an ecosystem (*Hazen et al., 2019*).

*G. galeus* was likely missed by the eDNA surveys because elasmobranchs are often underrepresented in eDNA analyses, as they may not shed high concentrations of DNA in the water column (*Walsh, Barrett & Hill, 2017*). Although *G. galeus*, was present in the reference libraries used in this study, elasmobranchs in general are frequently overlooked due to primer selection, which is often optimised for bony fish, or their poor representation in reference databases (*Pearce et al., 2021*; *Nishimura et al., 2022*). Tub gurnard (*C. lucerna*) and Yarrell's blenny (*C. ascanii*) were the other two species, along with tope (*G. galeus*), that were missed by the eDNA. Both *C. lucerna* and *G. galeus* are in the reference libraries used and were detected in the additional replicates taken. The reason they were missed in the first 29 replicates is likely due to a low percentage of eDNA present in the samples. This was also likely the case for *C. ascanii*, as it is in the reference library and this barcode matchers the primers used. These species were detected no more than two times on the BRUV, which would support the theory that their abundance was low in the environment at that time of the surveys or that they are rare species. Although the use of more metabarcoding primers or analysis of higher volumes of water filtered to extract eDNA from may have recovered these species, and we know that additional replicates did detect two of these species, using an alternative biomonitoring method such as BRUV surveys is highly recommended (*Cole et al., 2022*).

The results from our BRUV and eDNA surveys revealed that these methods target distinct species assemblages within Sussex Bay. BRUV and eDNA surveys demonstrated a significant difference in homogeneity of dispersions, suggesting that BRUV surveys had greater variability among sites than eDNA. This suggests that eDNA samples taken from different sites are more like one another, potentially due to the influence of tidal currents leading eDNA surveys to capture biodiversity over a broader spatial scale than BRUV surveys (*Deiner et al., 2016*; *Cole et al., 2022*). Highly dynamic marine conditions will influence the distance and direction in which eDNA travels from its point of origin to the point where the samples are taken (*Goldberg et al., 2016*). This could lead to more "mixing" of eDNA between different locations and thus resulting in false-positive species detections, and potentially in more homogeneous samples (*Jeunen et al., 2019a*). It is difficult to estimate the distance which eDNA travels due to the constant movement of water (*Thomsen & Willerslev, 2015*), while the presence of a species at a given site can be visually confirmed for BRUV surveys. The tides in the English Channel can be strong,

and are often stronger closer to the coastline, reaching up to 3.5 m/s in some instances (*Offshore Energy Strategic Environmental Assessment (SEA), 2009*). Thus, it is likely that there is substantial "mixing" of eDNA between sites with the result that our eDNA surveys may be providing an indication of the general species composition for Sussex Bay, rather than providing site-specific data. This may pose a challenge in the future when comparing sites within the trawling exclusion Byelaw zone and those outside of it. However, several studies have demonstrated that eDNA can identify discrete communities within marine habitats over short distances (*Thomsen et al., 2012*; *O'Donnell et al., 2017*; *Jeunen et al., 2019a*) and it may be that the lower variance for eDNA is a result of detecting 6 times more species per sample than the BRUV. Therefore, studies should be carried out to determine the spatial distribution of eDNA within coastal ecosystems such as Sussex Bay.

Despite our eDNA surveys proving to be more effective at detecting species richness than the BRUV surveys, one of the major limitations of eDNA metabarcoding is its current inability to determine life history traits such as sex ratios, age, body condition and behaviour (*Goldberg et al., 2016*; *Mynott, 2020*; *Gold et al., 2023*). These are all crucial metrics necessary for making informed management and conservation assessments. However, new advances in eDNA analysis are constantly emerging and it is likely that it will be possible to use eDNA to determine certain life history traits soon. Recently, for example, eRNA studies have been able to differentiate between signals originating from live or dead organisms (*Pochon et al., 2017*; *Wood et al., 2020*). Additionally, *Sigsgaard et al. (2020)* suggest that the use of epigenetic signals in eDNA data may be the future in determining the age and sex of species in a population. However, until these tools are further developed, we recommend the ongoing use of a combination of survey techniques. In this approach, eDNA can offer a comprehensive overview of the diversity present in an area, while traditional techniques, such as BRUV surveys, can provide more detailed insights about taxa and populations of interest (*Stat et al., 2019*).

Abundance is another measure which poses a challenge to eDNA surveys (*Sassoubre et al., 2016*; *Andruszkiewicz et al., 2017*; *Stoeckle, Soboleva & Charlop-Powers, 2017*). Certain studies have shown poor correlations between eDNA read counts and abundance (*Spear et al., 2015*; *Port et al., 2016*; *Yamamoto et al., 2016*; *Andruszkiewicz et al., 2017*; *Lamb et al., 2019*; *Yates, Fraser & Derry, 2019*; *Gold et al., 2023*). This could be due to many factors, including the direction and strength of currents, as well as methodological factors associated with metabarcoding including PCR bias (*Andruszkiewicz et al., 2019*). Fish breeding behaviour can also increase the amount of eDNA in the water (*Bylemans et al., 2017*). Additionally, these poor correlations may be attributed to a failure to consider the underlying mechanisms relating observed sequence read counts to the biology and biomass of the detected species (*McLaren, Willis & Callahan, 2019*; *Harrison, Sunday & Rogers, 2019*). Furthermore, the comparisons of eDNA concentrations to abundance are often estimated using traditional survey methods, which have their own biases (*Lyon et al., 2014*). For example, BRUV surveys come with several biases that can influence the accuracy of abundance estimates, *e.g.*, bait choice (*Jones et al., 2020*). Other studies have highlighted the biases of metagenomic sequencing and that the preferential measurement of some taxa over others leads to relative abundances of taxa being distorted from their
true values and incorrect conclusions being drawn about which taxa dominate different samples (*Brooks, 2016*; *Sinha et al., 2017*; *McLaren, Willis & Callahan, 2019*). Nevertheless, many recent studies have shown that eDNA can give a relatively accurate assessment of fish population abundance, especially when the study focus is on taxonomically similar and dimensionally comparable species (*Shelton et al., 2019*; *Shelton et al., 2022*; *Shelton et al., 2023*; *Di Muri et al., 2020*; *Hoshino et al., 2021*; *Spear et al., 2021*; *Russo et al., 2021*; *Stoeckle et al., 2021*; *Yates et al., 2021*; *Yates et al., 2023*; *Rourke et al., 2022*). Although using read counts as a proxy for species abundance was not in the scope of this article, we aim to explore this further in future research.

## Primer comparison

Both eDNA assays used in this study detected a distinct assemblage of marine vertebrates in Sussex Bay, which is consistent with previous research using multiple primers (*Nester et al., 2020*; *Cole et al., 2022*). *Nester et al. (2020)* and *Cole et al. (2022)* found that primers with shorter barcodes detected a greater species richness than those with longer barcodes. Shorter DNA fragments are more abundant in the environment (*Bylemans et al., 2017*), which may explain why the primers with shorter barcodes were found to be more sensitive in the studies mentioned above. Similarly, in our study, the MiFish 12S (140–200 bp) detected a higher number of species than the Valsecchi 16S assay (190–200 bp), which would align with this narrative. Nevertheless, a different community composition was detected by each assay, suggesting supplementary primer-template interactions and PCR efficiencies may also play a role in what was detected, and these results emphasise the importance of multi-assay eDNA metabarcoding for monitoring biodiversity (*Stat et al., 2017*; *Jeunen et al., 2019a*; *Nester et al., 2020*; *Cole et al., 2022*).

## eDNA sample replication

The analyses of the extra two eDNA samples taken at each site highlighted the importance of replication in eDNA biomonitoring studies, this aligned with the results of previous studies (*Andruszkiewicz et al., 2017*; *Harrison, Sunday & Rogers, 2019*; *Shelton et al., 2023*). The increased sampling effort captured the observed variability in eDNA signals between samples collected from a single site. This variability is likely due to the complexity of DNA signals and low eDNA concentrations within environmental samples (*Ficetola et al., 2015*). Although only seven more species were found with the two extra replicates, two of these species had been detected by the BRUV but not by the first eDNA replicate (*G. galeus* and *C. lucerna*). This further emphasises the benefit of replication for detecting more cryptic or rare species (*Beentjes et al., 2019*; *Stauffer et al., 2021*). The extra replicates also provided more informative data on community structure than a single sample per site. However, the results should be interpreted with caution as depth and treatment type are correlated with one another, with sites outside the trawling exclusion zone found further out from the coastline and therefore at greater depths. Nevertheless, our results align with the findings of previous studies, highlighting the ecological insight gained from replicates (*Doi et al., 2019*; *Macher et al., 2021*).

Had the MiFish 12S primer also been applied to the two extra eDNA replicates we would have found an even higher species richness than with the Valsecchi 16S primer alone. As

discussed above, the use of multiple primers can minimise the number of false negatives by targeting different genes with different amplification profiles, thus reducing the chance of poor amplification of a specific species (*Stat et al., 2017*; *Jeunen et al., 2019b*; *Alexander et al., 2020*). However, increasing the number of primers will increase the cost of eDNA analysis.

It is important to acknowledge that the expansion of sequence reference databases is essential to resolve assignments to a high taxonomic resolution (*West et al., 2020*). UK fish species, and notably the ones present in our study area are well represented in reference databases. However, there is a need for further expansion in understudied areas where the benefit of multiple primer pairs could also be greater (*Schenekar et al., 2020*).

One weakness of eDNA analysis that was encountered when carrying out this replication comparison was that not all our eDNA samples yielded results. There were only 21 out of our 29 sites for which all three replicate samples were successful preventing us from retrieving informative data.

## Cost and effort comparison

Overall, in the context of a 5-year timeframe, the BRUV surveys proved to have the lowest cost compared to both outsourced and in-house eDNA surveys. Nevertheless, eDNA analysis detected a higher species richness, resulting in a lower cost per species detected. However, it is important to caveat this finding by highlighting that although eDNA may detect a higher species richness, it likely encompasses species from a larger geographical range than the BRUV surveys, especially in areas with a high tidal range, such as Sussex Bay. Additionally, we found that conducting the eDNA analysis externally was the method which required the least time for the lead researchers. While the external eDNA analysis incurred a longer wait time for results (assuming researchers immediately begin processing samples) it can free up time for other research endeavours, albeit at a higher cost. Additionally, outsourcing the analysis may be a favourable option in specific circumstances or for researchers facing constraints, with regards to laboratory space or time to analyse the samples. Given the scarcity of both time and financial resources for researchers, the choice of method becomes subjective and depends on the researcher's specific objectives. We propose that a pragmatic approach involves combining both methods for programs aimed at surveying marine biodiversity over spatial and temporal scales. This integrated approach can help identify areas of high species richness and provide a comprehensive assessment of community structure and ecosystem health.

It is also important to note that when calculating the costs over 5 years of sampling we did not consider the effect of inflation or the likely decrease in cost of eDNA sequencing. Since the completion of the Human Genome Project, the cost of next-generation sequencing has decreased dramatically, outpacing Moore's Law (*Wetterstrand, 2021*). This is likely to be reflected in the price of outsourcing eDNA analysis and sequencing for biomonitoring projects in the coming years. Equally, BRUV technology is also likely to make great advancements in the coming years. The time to analyse BRUV footage varies depending on the number of organisms in the videos and the ease by which these can be identified, creating a significant bottleneck (*Cappo et al., 2003*; *Jäger et al., 2015*; *Sheaves et al., 2020*;

*Ditria et al., 2021*). Previous studies estimate the time spent on BRUV footage analysis to be double the length of the recorded video (*Willis & Babcock, 2000*; *Colton & Swearer, 2010*). However, with the rapid development of deep learning and associated AI tools used to automate or partially automate video analysis (*Marrable et al., 2022*; *Piechaud & Howell, 2022*), the time and effort used for BRUV footage analysis is likely to reduce considerably. Camera technology is also becoming more affordable and increasing in resolution quality, allowing more precise identification of population characteristics such as life stages, size and sex (*Whitmarsh, Huveneers & Fairweather, 2018*).

## CONCLUSIONS

We found that eDNA metabarcoding provides a more comprehensive view of marine vertebrate assemblage than BRUV, detecting 78 species *versus* 27 respectively. Importantly, eDNA detected the critically endangered European eel, *A. anguilla*. However, combining eDNA with the BRUV surveys allowed for a more holistic view of the marine vertebrate community, with the BRUV offering supplementary metrics including real-time data on habitat quality and the exact locations of the organisms identified. The investigation of the benefits of eDNA replication has shown that additional replicates can offer more informative results of community structure. Our study also found that BRUV surveys cost less overall, but that eDNA monitoring results in better value when accounting for the number of species detected. However, in contrast to BRUV surveys, the species detected by eDNA likely represent the species present within a larger geographical area than the specific site sampled, especially in areas with high tidal ranges, like our study site. Nevertheless, with enhanced time saving and greater conservation capabilities, eDNA surveys provide potential for greater alignment with SDG14, and the restoration goals set by the EU and Convention on Biological Diversity. However, until further advancements in eDNA technology allow for the detection of life history traits, using the two methods in conjunction may be essential depending on study goals. Our study has provided a monitoring baseline of fish and marine mammal community diversity for Sussex Bay upon introduction of the Nearshore Trawling Byelaw, allowing future monitoring studies to understand ecosystem structure in the coming years as the ecosystem recovers following removal of trawler fishing pressure. While the area of study was localised, findings presented herein are applicable to global aquatic biodiversity and conservation monitoring programs.

## ACKNOWLEDGEMENTS

We would like to thank Neville Blake: the captain of "New Dawn", the vessel on which we conducted our surveys, as well as his crew: Neil Frazer-Betts and Peter Everard and the University of Sussex masters and undergraduate students who assisted in the collection of data and video analysis. We would also like to acknowledge our reviewers and editors and the time they put into this manuscript.

## Funding

This work was supported by the Blue Marine Foundation, Sussex Wildlife Trust and NatureMetrics. The funders had no role in study design, data collection and analysis, decision to publish, or preparation of the manuscript.

## Grant Disclosures

The following grant information was disclosed by the authors:

The Blue Marine Foundation.

Sussex Wildlife Trust and NatureMetrics.

## Competing Interests

Nathan R. Geraldi works for NatureMetrics.

## Author Contributions

- Alice J. Clark performed the experiments, analyzed the data, prepared figures and/or tables, authored or reviewed drafts of the article, and approved the final draft.
- Sophie R. Atkinson performed the experiments, analyzed the data, authored or reviewed drafts of the article, and approved the final draft.
- Valentina Scarponi conceived and designed the experiments, performed the experiments, authored or reviewed drafts of the article, and approved the final draft.
- Tim Cane conceived and designed the experiments, performed the experiments, authored or reviewed drafts of the article, made the BRUV systems, and approved the final draft.
- Nathan R. Geraldi performed the experiments, authored or reviewed drafts of the article, and approved the final draft.
- Ian W. Hendy conceived and designed the experiments, authored or reviewed drafts of the article, and approved the final draft.
- J. Reuben Shipway conceived and designed the experiments, authored or reviewed drafts of the article, and approved the final draft.
- Mika Peck conceived and designed the experiments, performed the experiments, authored or reviewed drafts of the article, and approved the final draft.

## DNA Deposition

The following information was supplied regarding the deposition of DNA sequences:

The sequences are available at GenBank: PRJNA1092435.

## Data Availability

The raw data is available in the Supplemental Files.

## Supplemental Information

Supplemental information for this article can be found online at http://dx.doi.org/10.7717/peerj.17091#supplemental-information.

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
