# Peer review of "Cost-effort analysis of Baited Remote Underwater Video (BRUV) and environmental DNA (eDNA) in monitoring marine ecological communities"

_PeerJ, doi:10.7717/peerj.17091_

## Round 0.1 · original submission · Major Revisions

Dear Dr. Clark,

I have received three referees' reports on your paper “Cost-effort analysis of Baited Remote Underwater Video (BRUV) and environmental DNA (eDNA) in marine ecological community assessment recovery”. All reviewers believe that your work has the potential to make a relevant contribution to the discussion about marine biodiversity assessment methods; however, they also raise some concerns that need to be addressed before your paper can be considered for publication. Importantly, all reviewers raised concerns about the methods and required more clarification. They also highlight that some important references are missing and ask for a better interpretation of the figures and explanations of the patterns observed in their data sets. Finally, please try to include the time it took for the private company to do the laboratory and bioinformatic analyses in your cost and effort comparisons. On a personal note, I ask you to recognize the reviewer's effort in the acknowledgments section of your manuscript.

I hope you can attend to the referee's points, and I look forward to receiving your revised paper.

Best wishes

Guilherme

·

Basic reporting

Clear, unambiguous, professional English language used throughout.
Intro & background overall sufficient to show context. Literature well referenced & relevant but could be expanded a bit more.
Structure conforms to PeerJ standards.
Figures are relevant, high quality, well labelled & described but see required adaptation to figure that represents effort.
Raw data supplied (see PeerJ policy).

Experimental design

Original primary research within Scope of the journal.
Research question well defined, relevant & meaningful. It is stated how the research fills an identified knowledge gap.
Rigorous investigation performed to a high technical & ethical standard but see comments.
Methods were not fully described and clarification is needed.

Validity of the findings

See detailed comments below.

Additional comments

Overall, this paper was well written and includes interesting results. However, I have included some detailed comments below of things that need to be addressed in order to make this study replicable and findings to be more transparent.

Title: suggestion to invert order to “Cost-effort analysis of Baited Remote Underwater Video (BRUV) and environmental DNA (eDNA) in monitoring marine ecological communities”. The focus here was not really on the recovery, but rather on how these tools can be used for monitoring so I would remove that part from the title.

Abstract:
Line 49: change localized to regional

Introduction:
Lines 96-99: They can also be deployed in ecosystems that are hard to sample e.g. Shah Esmaeili et al. Comprehensive assessment of shallow surf zone fish biodiversity requires a combination of sampling methods. Marine Ecology Progress Series, v. 667, p. 131-144, 2021
Line 107, 110-111: also see ref mentioned above
Line 129-onward: It is important to mention here as well that one of the biggest limitations is the lack of references that are available in public databases leading to many species not being able to be identified to the species level (especially for certain regions of the world) or being misidentified. I know this is mentioned in the discussion but worth just bringing it up maybe.
Line 150: mention that you are looking at fish/vertebrate species, just pelagic/benthic leaves the impression that everything is considered

M&M:
Line 166: again, specify this is only fish/vertebrates
Line 205-206: Since there was also an interest in pelagic species, I am curious to know if the authors did any comparison of midwater/surface samples compared to benthic? The lack of DNA in the samples mentioned on lines 242-243 might be related to the lack of truly benthic species? Also, samples in more turbid waters or taken close to the sea floor can have the interference of the sediment and it is often recommended to use a cleanup step before PCR, not sure if this is in the methods by Alfaro-Cordova et al. 2022, but might be worth mentioning somewhere more explicitly.
Line 237: I understand these methods are written out in another paper, but I feel there should be at least some basic bioinformatic details mentioned here (QIIME,dada2,…).
Line 259-261: What do you mean by “worked”? The use of the eDNA samples is a bit unclear to me the way it has been written up here and in the text above. Please try to reformulate, it is weird to me that for all the sites, only the first replicates had enough DNA. A bit confusing.
Line 308-315: I have some issues with the way this effort was calculated… Most labs don’t have the resources to send samples for extraction and generally conduct steps up until sequencing. I understand this is reflected in the costs but it would be more fair to compare the effort for the entire process in the figure because it is sending a misleading message as is.

Discussion:
Lines 384-385: The methods only mentioned how fresh water species were removed, but not if there was some sort of quality control of species found using eDNA. Were all species that were detected known for the region? Were their occurrences reliable? Think this is important information to disclose.
Lines 394-402: Elasmobranchs might also be underrepresented due to the chosen primers or their representation in the reference libraries. Important to mention these things here as well.
Line 412: I highly recommend the bait eDNA should be removed from the analysis, since there is no way of making a distinction between naturally occurring and bait eDNA!
Line 438: I think abundance is as uncertain for BRUV as they are for eDNA. Although the MaxN method tries to limit the bias, unless you use extractive methods or individually identify organisms, even the BRUV has shortcomings here.
Line 478-481: That’s such a shame…
Line 489: I would argue that money is often more limited than time. It takes several weeks to get results back, because it actually takes a couple of weeks to extract, make libraries and sequence…
Line 513: this expression really gives the false impression…

Line 529: Acknowledgement of funding?

·

Basic reporting

This paper was written concisely with a clear introduction, methods, and conclusion. The authors compare two species diversity sampling methods: Baited Remote Underwater Video and eDNA barcoding. Individually, these methods were shown to have different strengths and weaknesses and when used in conjunction, paint a more transparent and holistic understanding of marine species biodiversity. The authors provided solid and sufficient background information backed up with adequate references. The figures were presented clearly. I would elaborate on the axes of Figure 4 and what the shapes of the BRUV and eDNA curves mean. I would explain the key differences between Figure 6 a and b; is there a significant difference between a and b? Figure 8 I would like to see standardization between outsourcing – consider doing a second figure to include outsourcing BRUV data analyses or doing eDNA in-house so you can directly compare the methods. Finally, consider adding another figure that complements Figure 8 showing a cost breakdown instead of an effort comparison.

Experimental design

The methods were carefully crafted to answer the authors’ hypotheses succinctly. The multiple hypotheses were presented in a straightforward form and were easily digestible. Adequate research planning and execution concisely answered the authors’ research questions. Referencing line 186, it appears that the researchers only used one vessel to complete the research. However, three BRUVs were simultaneously deployed; how is this possible if they were deployed 150m apart? Please clarify.

Validity of the findings

The authors did a commendable job explaining the comparisons between BRUV and eDNA, including the species diversity, the effort, and the cost analyses. While there were apparent differences and trade-offs between the methods, the authors conclude that these methods are best utilized in conjunction to gain a holistic understanding of the marine physical and biological environment. These data can be applied to future studies in the same region and serve as a baseline, and the implications of the results can be applied to future monitoring studies.

Additional comments

Line 74, I’d like to see more citations or examples here.
Lines 84-84, can you give example of biased or inefficient marine sampling
Lines 157-158 evidence or citation of that metric
Line 186 did you have multiple boats? How did you deploy simultaneously 150m apart?
Lines 242-244 rephrase out of passive voice to clarify meaning of sentence
Line 251 how many sites per treatment level
Line 283 write out which environmental variables
Line 292 explain why if you only do one eDNA, why you kept 3 BRUV per site
Line 386 please elaborate on the boat situation as such in line 186, would you need extra boats and crew
Line 391 contradicts the next sentence
Line 485 consider including an analysis that includes outsourcing BRUV data so you can make a direct comparison across BRUV and eDNA costs

Figure 8 – include a (the current one) and b (a new figure that shows outsourcing BRUV data processing).

Figure 9? Include a figure for the cost breakdown of each method.

Reviewer 3 ·

Basic reporting

The authors do not provide sufficient information within the manuscript on the bioinformatics conducted.

Experimental design

No comment.

Validity of the findings

Overall, the authors provide a very high level, cursory analysis of the data presented here and do not take substantial efforts to explain the patterns observed in their data sets. For example, the authors did not conduct a rigorous comparison of the two metabarcoding marker sets employed here, only focusing on the BRUV vs. aggregate eDNA comparisons. Such comparisons are warranted and would provide useful information for future eDNA monitoring efforts in the region. Likewise, 4 species were observed by BRUVs, but not identified by eDNA. The authors do not dig into the explanations for why this was the case. Is this due to missing reference barcodes? Is this due to limitations of taxonomic resolution within the marker set employed/ bioinformatics employed? Instead, the authors provide some speculative explanations that are loosely based on the literature. Furthermore, the authors failed to reference some important relevant research including Mirimin et al. 2021, West et al. 2020, Gold et al. 2023, and Cole et al. 2022 which all compared eDNA and BRUVS in various marine habitats. These papers have conducted very similar studies and thus should not only be referenced, but largely emulated and these studies have done much more due diligence in conducting thoughtful, detailed, and rigorous comparisons between approaches. The authors should spend more time diving into the “why” different approaches provided different observations of marine communities and taking better care to fit their findings within the broader context of the eDNA metabarcoding literature.
A glaring issue that I had with the manuscript was the cost and effort comparison between BRUVS and eDNA. The analysis conducted here is anything but an apples to apples comparison and is thus misleading at best. The overall ability to conduct a marine biodiversity survey is a function of cost and effort. Here the authors outsourced much of their eDNA analyses while conducitng the BRUVS analyses in house. In particular, the authors do not provide a fair comparison of labor costs and time needed to complete both analyses. Given that the labor and cost function has a surface of increasing costs and effort depending on outsourcing versus internalizing efforts, the authors need to provide a better accounting of all costs and time to truly do a fair comparison of both approaches. For example, for eDNA approaches, the authors did not account for the time it took for the private company to do the labwork and bioinformatic analyses. Likewise, the authors did not account for the cost of labor for the BRUVs video analysis. These glaring omissions make it impossible to accurately interpret the cost and effort comparisons presented here. The authors should take greater care to present a full accounting of the cost and effort of both approaches to provide a fair apples to apples comparison of both approaches or they should drop the comparisons altogether. Given that these comparisons in the peer reviewed literature may influence policy and management decisions, providing accurate information is paramount.

Additional comments

The authors present “Cost-effort analysis of Baited Remote Underwater 1 Video (BRUV) and environmental DNA (eDNA) in marine ecological community assessment recovery”. This study compares marine vertebrate biodiversity assessments between baited remote underwater video (BRUV) and eDNA metabarcoding off the Sussex Coast, UK. This study provides baseline characterization of marine fish communities prior to the enactment of a no-trawl zone needed for environmental impact assessment. This research both highlights the ability of eDNA metabarcoding approaches to detect a broader range of fish species than BRUVS while also identifying the shortcomings of using eDNA approaches lone. Overall, this work emphasizes the value of eDNA metabarcoding to provide meaningful fish assemblage information and indicates the potential for such approaches to enhance marine ecosystem monitoring efforts.

Although the paper has promise to provide valuable information to the broader eDNA metabarcoding, BRUV, and marine biodiversity assessment communities, I have significant reservations with the current form of this manuscript. Unless these concerns are adequately addressed in a future revision, I do not recommend this manuscript for publication.

Overall, the authors provide a very high level, cursory analysis of the data presented here and do not take substantial efforts to explain the patterns observed in their data sets. For example, the authors did not conduct a rigorous comparison of the two metabarcoding marker sets employed here, only focusing on the BRUV vs. aggregate eDNA comparisons. Such comparisons are warranted and would provide useful information for future eDNA monitoring efforts in the region. Likewise, 4 species were observed by BRUVs, but not identified by eDNA. The authors do not dig into the explanations for why this was the case. Is this due to missing reference barcodes? Is this due to limitations of taxonomic resolution within the marker set employed/ bioinformatics employed? Instead, the authors provide some speculative explanations that are loosely based on the literature. Furthermore, the authors failed to reference some important relevant research including Mirimin et al. 2021, West et al. 2020, Gold et al. 2023, and Cole et al. 2022 which all compared eDNA and BRUVS in various marine habitats. These papers have conducted very similar studies and thus should not only be referenced, but largely emulated and these studies have done much more due diligence in conducting thoughtful, detailed, and rigorous comparisons between approaches. The authors should spend more time diving into the “why” different approaches provided different observations of marine communities and taking better care to fit their findings within the broader context of the eDNA metabarcoding literature.
A glaring issue that I had with the manuscript was the cost and effort comparison between BRUVS and eDNA. The analysis conducted here is anything but an apples to apples comparison and is thus misleading at best. The overall ability to conduct a marine biodiversity survey is a function of cost and effort. Here the authors outsourced much of their eDNA analyses while conducitng the BRUVS analyses in house. In particular, the authors do not provide a fair comparison of labor costs and time needed to complete both analyses. Given that the labor and cost function has a surface of increasing costs and effort depending on outsourcing versus internalizing efforts, the authors need to provide a better accounting of all costs and time to truly do a fair comparison of both approaches. For example, for eDNA approaches, the authors did not account for the time it took for the private company to do the labwork and bioinformatic analyses. Likewise, the authors did not account for the cost of labor for the BRUVs video analysis. These glaring omissions make it impossible to accurately interpret the cost and effort comparisons presented here. The authors should take greater care to present a full accounting of the cost and effort of both approaches to provide a fair apples to apples comparison of both approaches or they should drop the comparisons altogether. Given that these comparisons in the peer reviewed literature may influence policy and management decisions, providing accurate information is paramount.
I also wanted to add that the authors only conduct presence/absence comparisons between the two methods with a small discussion alluding to the difficulty of obtaining accurate abundance estimates. However, recent work has demonstrated the advantages of the eDNA index (inverse Wisconsin double-standardization Kelly et al. 2019 and most recently implemented in Guri et al. 2023) in being able to improve differentiation of observed community composition while controlling for effects of amplification efficiency. Importantly, Guri et a. 2023 demonstrated how this transformation better explained known ecological patterns for species-specific differences. Similar approaches were used to compare BRUVs and eDNA in Gold et al. 2023. I strongly encourage the authors to incorporating this simple transformation to both the BRUV and eDNA data sets. Although the presence/absence comparisons conducted within are justified. The inclusion of abundance based comparisons would provide more insightful ecological comparisons while placing also addressing concerns raised by recent metabarcoding frameworks (Gloor et al. 2017, Silverman et al. 2021, Shelton et al. 2023, Gold et al. 2023).


Line Item edits:

Line 104 – I recommend inserting a new paragraph

Line 129 – I recommend inserting a new paragraph. I also recommend including additional references such as Goldberg et al. 2016

Line 146 – The authors should add a paragraph here that specifically discusses the previous comparisons of BRUVS and eDNA that have been conducted to provide context of previous results.

Line 239 – A brief description of the bioinformatics should be included here so the reader does not have to go back to Alfaro-Cordova et al. 2022 to get any information on how sequences were processed. Aside from using OTUs, it is unclear how taxonomic assignments were made. Best practices are to provide at least enough context to know the general bioinformatic schema employed.

Line 240 – I recommend referring to the locus as the vertebrate 16S or Valsecchi 16S gene region throughout as the MiFish primer set also detects vertebrates. The current nomenclature used throughout is confusing since both detect vertebrates and fish. I would recommend using MiFish 12S and Valsecchi 16S loci as the terminology employed.

Line 256 – . Please clarify how a total of 90 samples were obtained. I am not understanding how 90 BRUVs and 90 eDNA samples were collected since there were 29 sites total x 3

Line 241 –Please provide more information here. Was it a lack of DNA or a poor performing assay? See below for similar comment about the discussion.

Line 253 – Why didn’t the authors use the official UK National Tide Gauge Network data which is a more reputable data source: https://ntslf.org/data/uk-network-real-time ?

Line 262 – As discussed above, authors should consider using eDNA index as utilized in Kelly et al. 2019 which allows for comparisons of compositional abundances not just presence absence.

Line 264 – It’s interesting that the authors limited their comparisons to only species level assignments as many taxa can only be resolved to genus level using the marker sets employed. In Gold et al. 2023, some of the taxa that could only be resolved to species level by eDNA were the most abundant in BRUVs. The authors should include these comparisons and how they handled differing levels of taxonomic assignments between approaches in the manuscript or a supplemental results section.

Line 283 – What was the justification for applying a log transformation to the environmental data? Was this warranted to meet underlying assumptions of normality needed for the CCA? Please clarify and justify here or in the supplemental results.

Line 302 – Is the data for time to set up a BRUV recorded? The variance should be provided along with the average.

Line 308 – Please see the above comment about the issue with the cost and effort comparisons. This is not an apples to apples comparison since in this instance all of the effort was outsourced, making the effort comparisons difficult to interpret since it looks like the eDNA magically processed and annotated itself. Conducting laboratory work and bioinformatics requires a non-trivial amount of effort. Failing to account for that here in these comparisons because they were outsourced provides limited information and utility for other researchers who maybe deciding which biodiversity surveying tool to deploy in the future. I strongly recommend conducting a full accounting of time and cost for both BRUVs and eDNA if you wish to discuss this information in a rigorous fashion.

Line 324 – Please change to “eDNA alone”

Line 326 – Please include the standard deviation along with the mean.

Line 329 – The Shannon diversity index is not appropriate for presence absence data (See Herrera 1976) as the index assumes counts of individuals not semi-quantitative information on occurrences. I recommend utilizing Hill numbers as argued by Roswell et al. 2021, specifically reporting q=0 diversity order of Hill numbers. These statistics can be easily implemented using the iNEXT package (Hsieh et al. 2016) in R.

Line 333 – Please include the proportion of sites each species was detected.

Line 340 – Please report additional contextual information on the stress and associated statistics calculated from the CCA analysis. The current presentation of the results are incomplete.

Line 347 – This sentence is hard to understand. Please reword and clarify.

Line 350 – Please clarify what “it” is referring to?

Line 378 – Again these comparisons are not apples to apples. There is a surface of cost vs. effort that is a function of outsourcing versus doing work in house. You are showing two points along this surface relevant to your specific study and situation. These results are difficult to interpret without this context and the authors should take greater care to present an apples to apples comparisons accounting for the full time and cost of the eDNA analyses.

Line 387 – Authors conspicuously left out West et al. 2020, Cole et al. 2022, Gold et al. 2023, Mirimin et al. 2021 which conducted eDNA comparisons to BRUVs and seem highly relevant to this manuscript.

Line 392 – Do these species have reference barcodes in the database used to assign taxonomy? Likewise, can these taxa be resolved to species level resolution using the 2 employed biomarkers? You mentioned that there was low abundance of detection is that in reference to the unique number of times it was observed or the number of individuals observed? Providing both would be relevant and worth including in the discussion.

Line 404 – Here evenness is being used in a strange manner that is not the typical definition employed in ecology. Here you are saying evenness is how frequently a species is observed across sites as opposed to it’s relative abundance as defined in the Shannon index. The authors should follow conventional community ecology terminology and metrics to place this work better in the context of the field. I recommend following the conventions laid out in Roswell et al. 2021 and Hsieh et al. 2016.

Line 406 – As above, dominant species here is referencing the top three most occurring species which is an interesting use of the word dominant. However, since this was converted to presence/absence data it is not clear if these species are only frequently observed in low or high abundance. Again, please be more specific in your terminology and follow community ecology conventions.

Line 415 – Surprised that wasn’t thought of in the first place!

Line 420 – This can be specifically tested statistically using a homogeneity of dispersions test not just eye-balling the size of the clusters within the ordination. This is common practice in multivariate community ecology statistics and the omission of this is concerning. I strongly recommend that the authors consult a biostatistician with familiarity of both community ecology and compositional data sets (eDNA metabarcoding).

Line 423 – Your line of reasoning as for why eDNA fate and transport being impacted by tidal currents would decrease variability compared to BRUVS is unclear and not laid out in a coherent and detailed fashion.

Line 428 – Do you mean the “accuracy of eDNA”? Or are you actually referring to the source of the eDNA signature and how currents and tides may impact the ecological integration of the over space, time, and depth.

Line 430 – Why might this be? Please provide context here, are there different current/tidal regimes in these areas?

Line 434 – This paragraph does not provide a real meaningful discussion of the literature of ecological integration of eDNA being impacted by its fate and transport. The authors should provide a more meaningful discussion of how these results may impact ecological impact assessments and their results. Overall, the authors provide a very cursory discussion of their results, leaving a lot of room for more thoughtful and relevant discussion of their findings to the broader eDNA and marine biodiversity assessment fields.

Line 438 – This is not the first time this has been discussed or researched and thus should cite previous papers that have done these analyses. In general, this paragraph could be reorganized and presented in a more coherent fashion demonstrating more clearly the logical flow from one argument to the other.

Line 440 – The discussion of the relationship between abundance and eDNA is cursory at best. There is no mention of recent advancements in the field including McLaren et al. 2019, Shelton et al. 2023, Gold et al. 2023, Silverman et al. 2021, Andruszkiewicz et al. 2023, and Hoshino et al., 2021

Line 459 – But it wasn’t just 2 extra replicate right? It was 2 replicates across multiple sites? Please clarify appropriately.

Line 467 – Again this paragraph does not place these results in the context of the broader literature. We are not in 2015 when there was very little known about eDNA in marine ecosystems. There are dozens of papers that have demonstrated the value of replicate eDNA samples in marine ecosystems (e.g. Doi et al. 2019). These results are not surprising in the least and should be better couched in previous literature.

Line 473 – They don’t decrease the effect of amplification bias, but by targeting different genes with different assays with different amplification bias profiles, they increase the chance of avoiding poor amplification of a specific species. Please clarify the language here. This is a similar problem throughout. The authors should take better care to clarify the language and intention of each sentence.

Line 477 – This is a bit of a throw away sentence. The authors here should do their due diligence and determine the coverage of all the species in their region of interest or at least for all the species observed in the BRUVs data. The authors used a reference database with known taxonomic entries, doing the cross validation of the reference databases is not difficult and is expected. In general, the authors should spend more time diving deeper into the “why” of their results. Why did eDNA fail to detect 4 species observed in BRUVs? Is it due to reference database limitations? Is it due to under sampling? Many of these questions are answerable with the data in hand or at least more context for these results can be provided.

Line 480 – Is it 30 or 29 sites?

Line 481 – Why did this samples fail? Not enough DNA? Issues with PCR amplification? Inhibition? What tests were done on the samples to determine they did not have enough DNA? Providing more information and context here would provide readers a lot more meaningful and valuable information. Particularly as more and more eDNA samples are being conducted by private corporations it is important to understand the draw backs and limitations of outsourcing eDNA lab processing and analyses. This could be a meaningful discussion and addition to the literature if the authors took the time, care, and attention to detail to meaningfully discuss their experiences and findings.

Line 487 – You did not incorporate the cost of labor for the BRUV surveys? That is a glaring omission. As mentioned above, these cost comparisons are riddled with flaws. Providing meaningful cost comparisons is important as it may lead one agency or lab to use one technique over another. Providing misleading information and comparisons is concerning as these results may impact national, federal, state, academic, and NGO decision making.

Line 510 – “fish assemblages”. Please take care to catch all typos and grammatical errors throughout.

Figure 4 – The Y axis label is not abundance, this is occurrences. Please label figure legends appropriately throughout.

References


Andruszkiewicz Allan, Elizabeth, Ryan P. Kelly, Erin R. D'Agnese, Maya N. Garber‐Yonts, Megan R. Shaffer, Zachary J. Gold, and Andrew O. Shelton. "Quantifying impacts of an environmental intervention using environmental DNA." Ecological Applications: e2914.

Cole, Victoria J., David Harasti, Rose Lines, and Michael Stat. "Estuarine fishes associated with intertidal oyster reefs characterized using environmental DNA and baited remote underwater video." Environmental DNA 4, no. 1 (2022): 50-62.

Doi, Hideyuki, Keiichi Fukaya, Shin-ichiro Oka, Keiichi Sato, Michio Kondoh, and Masaki Miya. "Evaluation of detection probabilities at the water-filtering and initial PCR steps in environmental DNA metabarcoding using a multispecies site occupancy model." Scientific reports 9, no. 1 (2019): 3581.

Goldberg, Caren S., Cameron R. Turner, Kristy Deiner, Katy E. Klymus, Philip Francis Thomsen, Melanie A. Murphy, Stephen F. Spear et al. "Critical considerations for the application of environmental DNA methods to detect aquatic species." Methods in ecology and evolution 7, no. 11 (2016): 1299-1307.

Gold, Zachary, Andrew Olaf Shelton, Helen R. Casendino, Joe Duprey, Ramón Gallego, Amy Van Cise, Mary Fisher et al. "Signal and noise in metabarcoding data." Plos one 18, no. 5 (2023): e0285674.

Gold, Zachary, McKenzie Q. Koch, Nicholas K. Schooler, Kyle A. Emery, Jenifer E. Dugan, Robert J. Miller, Henry M. Page et al. "A comparison of biomonitoring methodologies for surf zone fish communities." Plos one 18, no. 6 (2023): e0260903.

Guri, G., Westgaard, J. I., Yoccoz, N., Wangensteen, O. S., Præbel, K., Ray, J. L., ... & Johansen, T. (2023). Maximizing sampling efficiency to detect differences in fish community composition using environmental DNA metabarcoding in subarctic fjords. Environmental DNA.

Gloor, G. B., Macklaim, J. M., Pawlowsky-Glahn, V., & Egozcue, J. J. (2017). Microbiome datasets are compositional: and this is not optional. Frontiers in microbiology, 8, 2224.

Herrera, Carlos M. "A trophic diversity index for presence-absence food data." Oecologia 25, no. 2 (1976): 187-191.
Hoshino, T., R. Nakao, H. Doi, and T. Minamoto. 2021. “Simultaneous Absolute Quantification and Sequencing of Fish Environmental DNA in a Mesocosm by Quantitative Sequencing Technique.” Scientific Reports 11: 1–9.

Hsieh, T. C., K. H. Ma, and Anne Chao. "iNEXT: an R package for rarefaction and extrapolation of species diversity (H ill numbers)." Methods in Ecology and Evolution 7, no. 12 (2016): 1451-1456.

Kelly, R. P., Shelton, A. O., & Gallego, R. (2019). Understanding PCR processes to draw meaningful conclusions from environmental DNA studies. Scientific reports, 9(1), 12133.

Mirimin, Luca, Sam Desmet, David López Romero, Sara Fernandez Fernandez, Dulaney L. Miller, Sebastian Mynott, Alejandro Gonzalez Brincau et al. "Don't catch me if you can–Using cabled observatories as multidisciplinary platforms for marine fish community monitoring: an in situ case study combining Underwater Video and environmental DNA data." Science of the Total Environment 773 (2021): 145351.

Roswell, Michael, Jonathan Dushoff, and Rachael Winfree. "A conceptual guide to measuring species diversity." Oikos 130, no. 3 (2021): 321-338.

Shelton, Andrew Olaf, Zachary J. Gold, Alexander J. Jensen, Erin D′ Agnese, Elizabeth Andruszkiewicz Allan, Amy Van Cise, Ramón Gallego et al. "Toward quantitative metabarcoding." Ecology 104, no. 2 (2023): e3906.
Silverman, J. D., Roche, K., Mukherjee, S., & David, L. A. (2020). Naught all zeros in sequence count data are the same. Computational and structural biotechnology journal, 18, 2789-2798.
Silverman, J. D., Bloom, R. J., Jiang, S., Durand, H. K., Dallow, E., Mukherjee, S., & David, L. A. (2021). Measuring and mitigating PCR bias in microbiota datasets. PLoS computational biology, 17(7), e1009113.

West, Katrina M., Michael Stat, Euan S. Harvey, Craig L. Skepper, Joseph D. DiBattista, Zoe T. Richards, Michael J. Travers, Stephen J. Newman, and Michael Bunce. "eDNA metabarcoding survey reveals fine‐scale coral reef community variation across a remote, tropical island ecosystem." Molecular ecology 29, no. 6 (2020): 1069-1086.

---

## Round 0.2 · Minor Revisions

Dear Dr. Clark,

I have received the comments on your revised paper from two referees. Both reviewers agree that your manuscript has significantly improved and I appreciate your - and your colleagues' - effort. However, the reviewers also raised some minor points that need to be addressed before your manuscript is accepted for publication.

I hope you can attend to the referee's points, and I look forward to receiving your revised paper.

Best wishes

Guilherme

·

Basic reporting

Clear writing and references in general in this new version. Do see comments below for more details.

Experimental design

Thank you for providing more detail on methods used.

Validity of the findings

In its current state, findings are more useful for anyone aiming to repeat this effort and/or make decisions on monitoring tools.

Additional comments

I believe the authors have made a significant effort in responding to the reviewers' comments and have addressed several of the issues that were raised. I do think there is still a bit of a lack in transparency in terms of the cost/effort, or better, the way this is presented. The cost estimation presented in the figure is very helpful but then is rebutted in the text by saying the cost/species detection is lower when using eDNA. We have to remember to be slightly sceptical about the number of species eDNA picks up in the light of circulation etc, so I feel like expressing the cost in $/species detected is not necessarily the best approach. I understand that authors are trying to make a case in favor of eDNA and I am a true believer of its power. But powerful tools need to be used with a necessary dose of caution. Comparing in-house and external analysis was a good addition but still does not express the effort (in hours) it takes to extract, make the libraries, sequence, run the bioinformatics and then have your relative abundance of species vs. watching the videos and obtaining these results more rapidly. There is also a different level of expertise required for both processes, which is something to be considered. Analyzing videos can be done by a technician/student with a basic level of species knowledge. The whole eDNA process, requires a combination of skill sets, including lab skills and bioinformatics, which are more specialized and have a steaper learning curve. Having used both methods myself, including the eDNA process from sample collection to end result (including all analytical steps) I believe this is something relevant to mention. Overall, I do believe this is a valuable study and can help monitoring programs in their decision making, it just needs to be a bit more honest/transparent. I believe you can still make your point without masking the truth.

·

Basic reporting

The authors did a commendable job of incorporating feedback, including clarifying methods, amending figures, and updating references. The language is primarily straightforward and clear. The added references provide for more clarification and legitimacy for their claims throughout the text. Further suggestions on additions and clarity are provided below.

Experimental design

On the second revision, the authors provided much more information on their processes and methods. Minor revisions are suggested for elucidation. The hypotheses are clearly stated and each step of the methods is more clearly explained.

Validity of the findings

This study appears to be a helpful contribution to the knowledgebase of ecological monitoring. In the second submission, most notably, the authors included a more comprehensive time, effort, and budget analysis of in house and outsourcing methods.

Additional comments

Line 201: replace semicolon with parenthesis
Line 206: lengthy sentence, consider breaking up, taking out words, or including punctuation
Line 213: add transitional word or connecting thought between these sentences
Line 221: consider moving this paragraph into the ‘study area’ section of materials and methods as it is your experimental site. The previous objectives feels like a wrap up of the introduction and a solid lead into your physical experiment.
Line 316: Was the Kemmerer sampler sanitized between deployments to avoid contamination between sites?
Line 235: teleost spelling
Line 331: how much DMSO? All other ingredients have quantities specified.
Line 334: Previous in-text citations have a comma after et al.
Line 335: comma after 69 C
Line 339 – 342: sentence is difficult to read as presented, consider rephrasing.
Line 324 – 343: font types are varied within these two paragraphs from Arial and Times, please consistently work with one font type
Line 346-347: can you have this one citation only at the end of the sentence?
Line 585 – 587: no data is still data – how can you rule out this significance? If the BRUV captured low counts of species, this could still be comparable data. The next sentence contradicts that this is experimental error therefore I do not believe you could logically exclude this site.
Line 599: comma after “estimates”
Line 664: what are the assumptions of normality specifically for a CCA?
Line 713 – 714: make into same paragraph
Line 731 – 736: Were these the top three species of each method, and were they present across both methods?
Line 741: what statistical test did you run to get these values? If it was a comparison of the two assays, did you not run a t test? Include degrees of freedom.
Line 939: consider a final sentence directly comparing the time effort comparison of the two methods.
Line 1013: remove the second “detected” in the sentence
Line 1033: condense this sentence to reduce the verbosity
Line 1162: comma after et al. for in text citation?
Line 1291: change researcher to researchers

Figures:
Consider adding captions for your figures.
Consider adding ellipses to appendix figure 2
Appendix Figure 3 and Appendix Figure 4 – change titles of both of these figures

---

## Round 0.3 · accepted · Accept

Dear Dr. Clark,

I have assessed the revised manuscript and believe it is ready for publication. Thank you for addressing all of the reviewers' comments.

Sincerely

Guilherme